# Active Surveillance for Carbapenem-Resistant *Enterobacterales* (CRE) Colonization and Clinical Course of CRE Colonization among Hospitalized Patients at a University Hospital in Thailand

**DOI:** 10.3390/antibiotics11101401

**Published:** 2022-10-13

**Authors:** Walaiporn Wangchinda, Visanu Thamlikitkul, Sureerat Watcharasuwanseree, Teerawit Tangkoskul

**Affiliations:** Division of Infectious Diseases and Tropical Medicine, Department of Medicine, Faculty of Medicine Siriraj Hospital, Mahidol University, Bangkok 10700, Thailand

**Keywords:** active surveillance, carbapenem-resistant *Enterobacterales*, clinical course, colonization

## Abstract

Optimal measures for preventing and controlling carbapenem-resistant *Enterobacterales* (CRE) depend on their burden. This prospective observational study investigated the prevalence and clinical course of CRE colonization in hospitalized patients at Siriraj Hospital, the largest university hospital in Thailand. Stool/rectal swab samples were collected from the patients upon admission, once weekly during hospitalization and every 1–3 months after discharge, to determine the presence of CRE in the stool. Between 2018 and 2021, a total of 528 patients were included. The prevalence of CRE colonization upon admission was 15.5%, while 28.3% of patients who tested negative for CRE on admission acquired CRE during their hospitalization. CRE colonization upon admission was usually associated with prior healthcare exposure. Among CRE-colonized patients, 4.7% developed a CRE clinical infection, with 60% mortality. No cutoff period that ensured that patients were free of CRE colonization in stool was identified, and isolation precautions should only be ceased if stool tests are negative for CRE. In conclusion, the prevalence of CRE colonization among hospitalized patients at Siriraj Hospital is high. CRE-colonized patients are at risk of developing subsequent CRE infection. To prevent CRE transmission within the hospital, patients at high risk of colonization should undergo CRE screening upon admission.

## 1. Introduction

Carbapenems are first-line agents for treating infections caused by third-generation cephalosporin-resistant *Enterobacterales*. With the massive use of carbapenems, the prevalence of infection and colonization with carbapenem-resistant *Enterobacterales* (CRE) has been increasing globally, especially in healthcare settings [1,2]. CRE infection is considered a severe threat to global health. Given the limited number of effective antibiotics against CRE, treatment of CRE infection is challenging and it is associated with high morbidity and mortality [3]. The burden of CRE colonization and infection varies among hospitals and patient populations [4]. CRE colonization and infection are more prevalent in patients with recent hospitalization or those administered broad-spectrum antibiotics, especially carbapenems [5,6]. Community spreading of CRE from patients with CRE colonization and infection is also a serious concern [7].

Intestinal colonization of multidrug-resistant *Enterobacterales* in hospitalized patients serves as a reservoir for disseminating *Enterobacterales* in the hospital setting. Recognizing CRE colonization in stool and providing infection prevention and control measures for patients with CRE colonization may prevent or contain CRE colonization and infection spread among hospitalized patients and hospital environments [4]. Moreover, selecting appropriate antibiotics to treat patients with CRE colonization who develop subsequent CRE infection should be very helpful for reducing morbidity and mortality. However, the benefits and cost-effectiveness of screening patients for CRE colonization upon hospital admission depend on the prevalence of CRE colonization in patients upon admission. In Thailand, a policy of active surveillance for CRE in all hospitalized patients upon admission has not been implemented in most hospitals due to limited microbiology laboratories and relevant resources. Therefore, data on the prevalence of CRE colonization upon admission and the clinical course of CRE colonization among hospitalized patients are essential to determine whether the active surveillance of CRE colonization should be implemented.

This study aimed to determine the prevalence of CRE colonization upon admission and the clinical course of CRE colonization among hospitalized patients at a university hospital in Thailand.

## 2. Results

### 2.1. Study Population

A total of 528 newly hospitalized patients were enrolled. Their mean age was 63.7 years, 56.3% were men, and the median hospital stay was 9 days. Most patients had chronic underlying diseases (95.8%) or previous healthcare exposure (81.3%). The median Karnofsky score was 70 [8]. A previous hospitalization within 3 months was observed in 46.8% of the patients, and 10.6% were transferred from a long-term care facility. Eleven patients (2.1%) had previous CRE colonization, with a median duration from the last detection of CRE colonization to the current admission of 74 days (range, 13–819 days). Six patients (1.1%) had a previous CRE infection before the current admission, with a median duration from the previous episode of CRE infection to the current admission of 42.5 days (range, 1–74 days). Two patients had a history of CRE colonization and infection before their current admission.

Most patients had received antibiotics within 3 months prior to the current admission (69.9%). Pulmonary diseases were the most common principal diagnosis for the current admission, followed by gastrointestinal/hepatic diseases and heart diseases. The all-cause mortality of the patients at hospital discharge was 16.7%.

### 2.2. Prevalence of CRE Colonization upon Admission

Of the 528 enrolled patients, the prevalence of CRE colonization upon admission to hospital in their stool or rectal swab samples was 15.5%. However, 2.8% of these patients with positive CRE detection upon admission had a history of CRE colonization or infection prior to admission. Therefore, the prevalence of newly detected CRE colonization among the patients upon admission was 12.6%.

### 2.3. Comparison of Patients with and without CRE Colonization upon Admission

Among the 82 patients with CRE colonization in their stool or rectal swab samples upon admission, 75.6% were colonized with carbapenem-resistant *K. pneumoniae*, 19.5% with carbapenem-resistant *E. coli*, and 4.9% with carbapenem-resistant *K. pneumoniae* and *E. coli*.

A detailed comparison of the 82 patients with CRE colonization and the 446 patients without CRE colonization upon admission by univariate analysis is given in Table 1. Patients with CRE colonization upon admission had a significantly lower Karnofsky score than those without CRE colonization. Neurological diseases; previous hospitalization, CRE colonization, or CRE infection; overall use of antibiotics within 3 months before admission; and use of antibiotics containing beta-lactams/beta-lactamase inhibitors, carbapenems, fluoroquinolones, vancomycin, fosfomycin, or colistin within 3 months before admission were significantly more prevalent in the patients with CRE colonization than in those without CRE colonization on admission. The use of a nasogastric tube, endotracheal/tracheostomy tube, or central venous catheterization was significantly more frequent in the patients with CRE colonization than in those without CRE colonization on admission. The median length of hospitalization of the patients with CRE colonization on admission was significantly longer than that of patients without CRE colonization on admission.

A comparison of the 82 patients with CRE colonization and the 446 patients without CRE colonization upon admission by multivariate analysis revealed that previous hospitalization within 3 months, previous use of beta-lactams/beta-lactamase inhibitors or carbapenems, previous CRE colonization, and endotracheal/tracheostomy tube insertion prior to admission were associated with CRE colonization upon admission (Table 2).

Among the 82 patients with CRE colonization upon admission, 86.6% had CRE colonization up to the last stool or rectal swab samples that were collected prior to discharge. The median duration of CRE colonization from the first detection of CRE to hospital discharge was 8 days (range, 1–49 days). Therefore, 13.4% of the patients with CRE colonization upon admission did not have CRE colonization in their follow-up stool or rectal swab samples before their hospital discharge. The median CRE colonization from the first detection of CRE to the absence of CRE in the patients with CRE colonization upon admission was 6 days (range, 4–26 days).

### 2.4. Acquisition of CRE during Hospitalization

Among 233 patients who had no CRE upon admission and had at least one follow-up stool or rectal swab test for CRE, 66 of them had positive CRE in their stool or rectal swab samples. Therefore, 28.3% of patients acquired CRE during their hospitalization. The median duration from admission to the presence of CRE colonization in the 66 patients who developed CRE colonization during their hospitalization was 12 days (range, 5–52 days).

### 2.5. Comparison of Patients with and without CRE Acquisition during Hospitalization among Patients without CRE Colonization upon Admission

Of the 233 patients without CRE colonization upon admission who had follow-up stool or rectal swab samples for CRE detection, 28.3% developed CRE colonization in their stool or rectal swab samples. Of these, 83.3% were colonized with carbapenem-resistant *K. pneumoniae*, 13.6% with carbapenem-resistant *E. coli*, and 3.0% with carbapenem-resistant *K. pneumoniae* and *E. coli*.

The results of a univariate analysis of the 66 patients with CRE colonization and the 167 patients without CRE acquisition during hospitalization are summarized in Table 3. The patients with CRE colonization received central venous catheterization or underwent surgery significantly more frequently than those without CRE colonization during admission. Respiratory tract infection and gastrointestinal/hepatobiliary tract infection were significantly more common among the patients with CRE colonization than among those without CRE acquisition during hospitalization. The median length of hospital stay of the patients with CRE colonization was significantly longer than that of patients without CRE colonization.

Multivariate analyses of the 66 patients with CRE colonization and the 167 patients without CRE acquisition during hospitalization revealed that undergoing surgery, respiratory tract infection, and gastrointestinal/hepatobiliary tract infection were associated with CRE acquisition during hospitalization. The patients with CRE acquisition during hospitalization had a significantly longer hospital stay than those without CRE acquisition during hospitalization (Table 4).

Among the 66 cases of CRE acquisition during hospitalization, 81.8% had CRE colonization up to and including discharge. The median duration of CRE colonization from the first detection of CRE to hospital discharge was 12 days (range, 5–52 days). The absence of CRE colonization in follow-up stool or rectal swab samples in the last CRE detection test before discharge was observed in only 18.2% of the patients with CRE acquisition during hospitalization. The median duration of CRE colonization from the first detection of CRE to the absence of CRE in the patients with CRE acquisition during hospitalization was 8 days (range, 5–51 days).

### 2.6. Prevalence of CRE Fecal Carriage upon Admission and during Hospitalization Classified by Year during 2018–2021

The prevalence of CRE colonization in the stool samples of the enrolled patients in 2018, 2019, 2020, and 2021 is shown in Table 5. There was a trend of increasing prevalence of CRE colonization over the 4 years of the study, with the exception of 2020, when the COVID-19 pandemic occurred and the number of patients was small, because it was quite difficult to enroll patients during the COVID-19 pandemic. Although COVID-19 was still epidemic in our hospital in 2021, the severity of the disease was much lower than that in 2021 and most of the patients and health personnel had received COVID-19 vaccination. Therefore, the number of enrolled patients in 2021 was greater than that in 2020.

### 2.7. Development of CRE Infection

Ten patients developed CRE infection during hospitalization. Therefore, the rate of CRE infection was 1.9% of the total enrolled patients. However, the overall rate of CRE infection was 4.7% among 148 patients with CRE colonization upon admission and during hospitalization. All patients with CRE infection had a prolonged course of current or previous hospitalization. The median duration from the first detection of CRE colonization in stool or rectal swab to CRE infection was 15 days. The rate of CRE infection in the patients with CRE colonization upon admission was 3.7% among 82, and the rate of CRE infection in the patients with CRE acquisition during hospitalization was 6.1% among 66 patients.

Most CRE infections were caused by *K. pneumoniae* (80%), followed by *E. coli* (10%) and *Enterobacter* spp. (10%). The most common antibiotic regimens for treating these CRE infections were colistin alone and colistin in conjunction with fosfomycin or meropenem. After treatment completion, 70% of the patients with CRE infections who had follow-up stool or rectal swab samples for CRE detection had persistent carbapenem-resistant *K. pneumoniae* colonization. All patients with CRE infection had multiple comorbidities. Two of them died from complications related to their comorbidities, which was not directly related to CRE infection. The in-hospital mortality of the patients with CRE infection was 60% and the CRE-related mortality was 40%.

### 2.8. CRE Colonization after Hospital Discharge

Of the 148 patients with CRE colonization upon admission and during hospitalization, 111 patients (75%) were discharged alive from the hospital. However, only 41 of the 111 patients (36.9%) collected and submitted at least one stool sample from home to enable the investigators to determine the continued presence of CRE. The median follow-up period from hospital discharge was 116.5 days (range, 23–303 days). Stool samples became negative for CRE in 78.0% of patients in up to 303 days. The median duration of colonization (from initial detection until negative CRE in stool) in patients with negative CRE in stool samples sent from home was 85 days (range, 20–188 days). Nine of the 41 patients (22.0%) had persistent CRE colonization at their last follow-up. The median duration of CRE colonization in patients with persistent CRE colonization in their stool from hospital discharge to their last follow-up at home was 77 days (range, 35–303 days).

Among the 380 patients without CRE colonization upon admission and during hospitalization, 329 patients (86.6%) were discharged alive from the hospital. Of those, 28.3% collected and submitted at least one stool sample from home to enable the investigators to determine the presence of CRE. Eight patients (8.6%) developed CRE colonization after their discharge from hospital. None of these patients had never had CRE colonization upon admission or during hospitalization. The median duration of developing CRE colonization in the stool after hospital discharge was 46.5 days (range, 39–78 days).

## 3. Discussion

Epidemiological data on CRE colonization and infection among hospitalized patients in Thailand are limited. The data from previous studies usually described the microbiological aspects of CRE isolates. However, the studies did not systematically collect clinical samples from patients with CRE, clinical data related to prevalence and acquisition rate, or relevant clinical information on patients with CRE colonization and infection [9,10,11,12]. However, a cross-sectional study on the prevalence of CRE among intensive care unit patients in Thailand in 2018 determined a prevalence of CRE colonization of 3.2% [13].

This is the first study in Thailand that performed active surveillance to (1) determine the prevalence of CRE colonization and (2) analyze patients’ clinical courses from admission to general medicine wards at Siriraj Hospital until after discharge from hospital. We could not enroll all newly hospitalized patients. Although there were many new patients each day, we had insufficient resources to study all patients from admission to hospital discharge and later. Therefore, we had to enroll only patients admitted to general medicine wards on random days. A stool or rectal swab sample was collected from each patient to determine the presence of CRE, as the bacteria usually colonize the gastrointestinal tract [14,15]. This study used selective media (CRE CHROMagar) to detect CRE in the stool or rectal swab samples. The method is a simple and reliable means of detecting CRE in clinical samples containing many types of bacteria, such as stool [16,17]. Moreover, the selective agar test results on the presence of CRE in clinical samples are available the day after samples are collected.

We collected the samples from the patients upon admission, during hospitalization, and after discharge from the hospital. We aimed to determine the (1) prevalence of CRE colonization upon admission, (2) CRE acquisition rate during hospitalization, (3) rate of CRE infection in patients with CRE colonization, and (4) clinical course of CRE colonization and infection.

The prevalence of CRE colonization in the stool or rectal swab samples among the 528 patients admitted to general medicine wards in this study was 15.5%. Such prevalence is high compared with data from previous reports in Thailand [9,11]. However, some patients with CRE colonization upon admission were known to have had CRE colonization or infection before their current hospitalization. However, the presence of CRE colonization upon admission of the patients without a prior history of CRE colonization or infection was still prevalent. Their CRE colonization should be associated with healthcare exposures, since many patients had chronic underlying diseases, previous hospitalization, and antibiotic use, and they underwent medical procedures, similar to previous studies [18,19,20]. However, CRE colonization upon admission of such patients could be due to the transmission of CRE from other people or patients who are CRE carriers [21,22]. It is unlikely that these patients would have acquired CRE from the community, since the presence of CRE in the general community in Thailand is still quite low [23,24,25,26].

In the current investigation, the prevalence of CRE colonization in the stool or rectal swab samples of the patients upon admission to general medicine wards was rather high. Without active surveillance for CRE colonization among individuals about to be hospitalized, patients with CRE colonization may transmit CRE to other patients, health personnel, and the hospital environment. If individuals who are to be hospitalized are known to have CRE colonization, they could be isolated to prevent CRE transmission.

Unfortunately, implementing active CRE surveillance for all patients being hospitalized in Thailand has potentially substantial resource implications. On the one hand, it is true that detecting CRE in stool samples is not expensive. Nevertheless, information on the presence or absence of CRE colonization may not be sufficient to warrant a policy of active surveillance of CRE in all patients before their hospitalization. This is because many facilities (such as isolation rooms and personal protective equipment) would be needed to ascertain the patients’ CRE colonization status. A cost-effectiveness analysis would first be necessary to examine the impact of providing universal CRE determination and isolation precautions for patients with CRE colonization prior to admission. The extent of CRE transmission from patients with CRE colonization to others also needs to be established.

An alternative approach is that the hospital could consider restricting the screening of CRE colonization to only high-risk patients upon admission. Such screening would be based on the clinical parameters that our multivariate analyses identified as associated with CRE colonization upon admission. These factors are previous hospitalization, previous use of beta-lactam/beta-lactamase inhibitors or carbapenems, previous CRE colonization, and endotracheal tube or tracheostomy tube insertion before admission. It should be emphasized that the high prevalence of CRE colonization among the patients upon admission was observed in individuals admitted to general medicine wards. Data on the prevalence of CRE colonization in patients admitted to other types of wards (e.g., surgery, obstetrics and gynecology, and ophthalmology) are not available. However, it is speculated that the prevalence of CRE fecal carriage in patients hospitalized in other departments should be less than that of the Department of Medicine, because those patients have fewer healthcare-associated conditions, such as having chronic underlying diseases, prior use of antibiotics, and receipt of antibiotics during hospitalization. Therefore, a study on the active surveillance of CRE in patients hospitalized in other departments, in addition to the Department of Medicine, should be conducted to consider whether the active surveillance of CRE should be performed in all hospitalized patients or only in some patients in some departments.

The rate of CRE acquisition during hospitalization (28.3%) exceeded the 15.5% prevalence of colonization upon admission. The higher value could result from the many drivers for the acquisition of CRE colonization during hospitalization. The relatively high rate of CRE acquisition during hospitalization suggests that a universal infection prevention and control measures and the appropriate use of antibiotics should be enforced and reinforced. An alternative measure, which would need substantial resources, is to screen for CRE in all hospitalized patients every 1 to 2 weeks and employ contact precaution measures for patients who are CRE colonizers. It should be kept in mind that most patients with CRE colonization had persistent CRE colonization during hospitalization and after discharge.

This study also revealed that some patients who did not have CRE colonization upon admission and during hospitalization developed CRE colonization after discharge from the hospital. The reasons for this observation are unclear. It could result from a suppression effect of the anti-CRE antibiotics administered to the patients during their stay in the hospital. It is also possible that there were false-negative results for CRE from the patients’ previous surveillance cultures undertaken while they were still hospitalized. Additionally, the patients might have had new healthcare exposures or acquired CRE from other sources after they returned to their homes. This observation is worrisome because infection prevention and control measures will not be applied to these patients, since the relevant health personnel will not be aware that these patients are CRE colonizers if the detection of CRE in the patients’ stool samples is not performed. A larger study to determine the prevalence of the new acquisition of CRE and its associated factors in patients who do not have CRE acquisition during hospitalization but develop CRE colonization after discharge from hospital should be conducted.

Patients with CRE colonization are at risk of developing CRE infection and transmitting CRE to other patients, healthcare personnel, the hospital environment, their family members and partners, and the community environment. Subsequent CRE infections following CRE colonization by the same CRE phenotypic bacteria have been observed in previous studies [27,28,29,30], especially among neutropenic patients, other immunocompromised patients, or critically ill patients. The translocation of bacteria from the intestine to blood is considered a potential mechanism. Infections caused by CRE were associated with high morbidity and mortality due to the limitation of treatment options. The present study found that CRE infection developed in 4.7% of the patients with CRE colonization, with 60% mortality. Furthermore, most patients had persistent CRE colonization in their gastrointestinal tract at discharge from the hospital. The rate of CRE infection among patients with CRE colonization upon admission and during hospitalization in the present study was less than the 16.5% reported in a meta-analysis by McConville and colleagues [31]. In our study, we did not know the extent to which CRE colonization was transmitted to other inpatients or outpatients. This is because we did not perform CRE screening of patients in contact with those with CRE colonization.

The most common CRE species in this study was *K. pneumoniae*, similar to the results of other reports [32]. This observation would explain why 80% of the CRE infections in the patients with CRE colonization in our study were caused by *K. pneumoniae*.

The duration of CRE colonization in the patients in this study was variable. We observed that some patients had persistent CRE colonization for a protracted period, even after discharge from the hospital. Consequently, it is difficult to specify a cutoff period within which CRE should disappear from CRE colonizers. The implication is that the detection of CRE in stool samples should be performed. This approach will ensure that a patient is free of CRE and signal when contact precautions can be discontinued. However, it should be kept in mind that patients who are free of CRE colonization can still develop CRE colonization. For example, they may have healthcare exposures (such as receiving broad-spectrum antibiotics) or receive CRE from other patients, healthcare personnel, or the hospital environment.

In conclusion, CRE colonization among hospitalized patients upon admission and during hospitalization was prevalent in the medicine wards of Siriraj Hospital, and it was usually associated with healthcare exposures. Patients at high risk of CRE colonization might need CRE screening upon admission to the hospital so that timely infection prevention and control measures can be implemented. Some patients with CRE colonization developed subsequent CRE infection. The duration of CRE colonization was highly variable. Unfortunately, no cutoff period could be established to ensure that CRE-colonized patients are free of CRE and signal when contact precautions can be discontinued. Consequently, confirmation of the absence of CRE by laboratory detection of CRE is necessary.

## 4. Patients and Methods

This prospective observational cohort study was approved by the Institutional Review Board of the Faculty of Medicine, Siriraj Hospital. It was conducted between November 2018 and November 2021 at Siriraj Hospital, a 2300-bed tertiary-care university hospital in Bangkok, Thailand. Written informed consent was obtained from all enrolled patients.

Adult patients admitted to general medicine wards on randomly selected surveillance days were enrolled. The once-weekly random date was scheduled because the study team had limited time and resources to enroll all patients. Stool or rectal swab samples were obtained from all enrolled patients as soon as possible but within 72 h of hospitalization. The aim was to determine the prevalence of CRE colonization in the stool of patients upon admission to general medicine wards. Stool or rectal swab samples were subsequently collected for CRE detection once weekly throughout the patients’ hospital stays. The purpose was to determine (1) the rate of CRE acquisition in hospitalized patients’ stool and (2) the clinical course of CRE colonization during hospitalization of patients with and without CRE colonization in their stool upon admission.

After hospital discharge, all survived participating patients and/or their relatives at hospital discharge were asked and instructed to collect stool samples at home every 1 to 3 months. The samples were sent to the investigators to determine the presence of CRE. The stool collection and examination process were repeated for each patient until samples were negative for CRE. CRE was detected in the collected stool or rectal swab samples using chromogenic selective agar (CHROMagar mSuperCARBA, CHROMagar Company, Paris, France). The presence of CRE in stool samples was detected by the growth of blue or pink bacterial colonies on the agar. Identification of CRE species was performed by manual biochemical tests, including the triple sugar iron test, lysine iron agar slants test, indole test, motility test, ornithine decarboxylase test, urease test, and malonate test, which were locally prepared in the laboratory using the purchased reagents/materials (Oxoid Ltd.; Hampshire, UK or BBL/Difco Diagnostic, Becton Dickinson; Sparks, MD, USA), and the oxidase test (BBL/Difco Diagnostic, Becton Dickinson; Sparks, MD, USA), to identify *E. coli* and *Klebsiella* species.

### Data Analysis

Data were analyzed with PASW Statistics for Windows, version 16.0 (SPSS Inc, Chicago, IL, USA). The variables were displayed as descriptive statistics and were compared using Fisher’s exact test, chi-squared test, or Student’s *t*-test, as appropriate. A probability (*p*) value ≤ 0.05 was considered statistically significant. Multivariate analysis was performed using a backward stepwise binary logistic regression method, with CRE colonization as the dependent variable and the potential risk factors, such as previous use of antibiotics, previous CRE colonization or infection, and duration of hospitalization, for the development of CRE colonization as the independent variables.

## Figures and Tables

**Table 1 antibiotics-11-01401-t001:** Comparison of 528 patients with and without CRE colonization on admission.

Variables	CRE Colonization (N = 82)	No CRE Colonization (N = 446)	*p* Value
Male	44 (53.7%)	253 (56.7%)	0.61
Mean age (SD), years	67.0 (17.2)	63.1 (18.1)	0.07
Median Karnofsky score (range)	60 (20–100)	70 (20–100)	<0.001
Comorbid conditions	78 (95.1%)	428 (96.0%)	0.76
Hypertension	52 (63.4%)	272 (61.0%)	0.68
Diabetes mellitus	29 (35.4%)	176 (39.5%)	0.48
Cardiovascular diseases	28 (34.1%)	152 (34.1%)	0.99
Renal diseases	26 (31.7%)	139 (31.2%)	0.92
Neurological diseases	33 (40.2%)	113 (25.3%)	0.006
Gastrointestinal/hepatobiliary diseases	19 (23.2%)	89 (20.0%)	0.51
Pulmonary diseases	13 (15.9%)	83 (18.6%)	0.55
Solid malignancy	14 (17.1%)	79 (17.7%)	0.89
Autoimmune diseases	12 (14.6%)	62 (13.9%)	0.86
Hematological malignancy	6 (7.3%)	15 (3.4%)	0.12
Solid organ transplant	2 (2.4%)	6 (1.3%)	0.62
Previous healthcare exposure			
Medical procedure within 3 months	55 (67.1%)	252 (56.5%)	0.08
Indwelling urinary catheter	48 (58.5%)	220 (49.3%)	0.13
Nasogastric tube	35 (42.7%)	104 (23.3%)	<0.001
Endotracheal tube/tracheostomy tube	19 (23.2%)	39 (8.7%)	<0.001
Chronic dialysis	6 (7.3%)	28 (6.3%)	0.73
Central venous catheterization	10 (12.2%)	16 (3.6%)	0.003
Previous hospitalization within 3 months	59 (72.0%)	188 (42.2%)	<0.001
Use of immunosuppressive agents within 3 months	19 (23.2%)	100 (22.4%)	0.88
Transfer from long-term care facility	13 (15.9%)	43 (9.6%)	0.09
Major surgery within 3 months	6 (7.3%)	18 (4.0%)	0.15
Use of antibiotics within 3 months	71 (86.6%)	298 (66.8%)	<0.001
Cephalosporins	36 (43.9%)	191 (42.8%)	0.86
Beta-lactams/beta-lactamase inhibitors	47 (57.3%)	127 (28.5%)	<0.001
Macrolides	9 (11.0%)	81 (18.2%)	0.11
Fluoroquinolones	16 (19.5%)	48 (10.8%)	0.03
Carbapenems	21 (25.6%)	31 (7.0%)	<0.001
Penicillins	6 (7.3%)	26 (5.8%)	0.62
Vancomycin	11 (13.4%)	10 (2.2%)	<0.001
Fosfomycin	6 (7.3%)	2 (0.4%)	<0.001
Aminoglycosides	2 (2.4%)	4 (0.9%)	0.24
Colistin	2 (2.4%)	0 (0.0%)	0.02
Previous CRE colonization	11 (13.4%)	4 (0.9%)	<0.001
Previous CRE infection	6 (7.3%)	0 (0.0%)	<0.001
Duration in observation room prior to hospitalization (range), days	1 (0–5)	0 (0–9)	0.13
Median length of hospitalization (range), days	12.5 (2–68)	9 (1–177)	0.008
All-cause mortality at hospital discharge	19 (23.2%)	69 (15.5%)	0.09

Abbreviation: CRE, carbapenem-resistant *Enterobacterales*.

**Table 2 antibiotics-11-01401-t002:** Multivariate analyses of factors associated with CRE colonization on admission to hospital.

Variables	Univariate Analysis OR (95% CI; *p* Value)	Multivariate Analysis OR (95% CI; *p* Value)
Median Karnofsky score (range)	0.97 (0.96–0.99; *p* < 0.001)	
Neurological diseases	1.99 (1.22–3.24; *p* = 0.006)	
Previous hospitalization within 3 months	3.52 (2.10–5.91; *p* < 0.001)	1.93 (1.06–3.49; *p* = 0.03)
Use of beta-lactams/beta-lactamase inhibitors within 3 months	3.37 (2.08–5.47; *p* < 0.001)	2.10 (1.20–3.66; *p* = 0.009)
Use of carbapenems within 3 months	4.609 (2.49–8.53; *p* < 0.001)	2.63 (1.29–5.35; *p* = 0.008)
Use of fluoroquinolones within 3 months	2.01 (1.08–3.75; *p* = 0.03)	
Use of vancomycin within 3 months	6.76 (2.77–16.49; *p* < 0.001)	
Use of fosfomycin within 3 months	17.53 (3.47–88.45; *p* = 0.001)	
Previous CRE colonization	17.12 (5.31–55.24; *p* < 0.001)	6.95 (1.99–24.26; *p* = 0.002)
Previous use of nasogastric tube	2.45 (1.50–4.00; *p* < 0.001)	
Previous use of endotracheal tube/tracheostomy tube	3.15 (1.71–5.79; *p* < 0.001)	2.49 (1.26–4.90; *p* = 0.008)
Previous use of central venous catheterization	3.73 (1.63–8.55; *p* = 0.002)	

Abbreviation: CRE, carbapenem-resistant *Enterobacterales*.

**Table 3 antibiotics-11-01401-t003:** Comparison of 233 patients with and without CRE acquisition during hospitalization.

Variables	CRE Colonization (N = 66)	No CRE Colonization (N = 167)	*p* Value
Antibiotic use during hospitalization			
Beta-lactams/beta-lactamase inhibitors	39 (59.1%)	87 (52.1%)	0.33
Cephalosporins	31 (47.0%)	81 (48.5%)	0.84
Carbapenems	22 (33.3%)	47 (28.1%)	0.43
Fluoroquinolones	14 (21.2%)	33 (19.8%)	0.80
Vancomycin	13 (19.7%)	22 (13.2%)	0.21
Macrolide	10 (15.2%)	18 (10.8%)	0.36
Colistin	6 (9.1%)	9 (5.4%)	0.37
Penicillins	5 (7.6%)	11 (6.6%)	0.78
Aminoglycosides	3 (4.5%)	2 (1.2%)	0.14
Fosfomycin	1 (1.5%)	3 (1.8%)	1.00
Procedure during hospitalization			
Indwelling urinary catheter	49 (74.2%)	117 (70.1%)	0.53
Mechanical ventilator	28 (42.4%)	60 (35.9%)	0.36
Central venous catheterization	20 (30.3%)	28 (16.8%)	0.02
Surgery	13 (19.7%)	17 (10.2%)	0.05
Principal diagnosis in this admission			
Pulmonary diseases	18 (27.3%)	46 (27.5%)	0.97
Gastrointestinal/hepatic diseases	9 (13.6%)	28 (16.8%)	0.56
Heart diseases	9 (13.6%)	21 (12.6%)	0.83
Renal diseases	6 (9.1%)	18 (10.8%)	0.70
Cerebrovascular diseases	9 (13.6%)	16 (9.6%)	0.37
Solid malignancy	5 (7.6%)	12 (7.2%)	1.00
Autoimmune diseases	2 (3.0%)	9 (5.4%)	0.73
Hematological malignancy	2 (9.0%)	8 (4.8%)	0.73
Diagnosis of infection during hospitalization			
Respiratory tract infection	37 (56.1%)	69 (41.3%)	0.04
Urinary tract infection	16 (24.2%)	25 (15.1%)	0.10
Gastrointestinal/hepatobiliary tract infection	15 (22.7%)	17 (10.2%)	0.01
Bloodstream infection	7 (10.6%)	22 (13.2%)	0.59
Skin and soft tissue infection	5 (7.6%)	23 (13.8%)	0.19
Central nervous system infection	2 (3.0%)	4 (2.4%)	0.68
Cardiovascular system infection	1 (1.5%)	0 (0.0%)	0.29
Median length of hospitalization (range), days	23.5 (3–177)	13 (2–71)	<0.001
All-cause mortality at hospital discharge	18 (27.3%)	33 (19.8%)	0.21

**Table 4 antibiotics-11-01401-t004:** Multivariate analyses of factors associated with CRE acquisition during hospitalization.

Variables	Univariate Analysis OR (95% CI; *p* Value)	Multivariate Analysis OR (95% CI; *p* Value)
Central venous catheterization	2.16 (1.11–4.19; *p* = 0.02)	
Surgery	2.16 (0.99–4.76; *p* = 0.06)	2.153 (0.95–4.87; *p* = 0.07)
Respiratory tract infection	1.81 (1.02–3.22; *p* = 0.04)	2.05 (1.12–3.74; *p* = 0.02)
Gastrointestinal/hepatobiliary tract infection	2.60 (1.21–5.57; *p* = 0.01)	2.99 (1.34–6.65; *p* = 0.007)

**Table 5 antibiotics-11-01401-t005:** Prevalence of CRE fecal carriage of the enrolled patients upon admission and during hospitalization.

	Nov. to Dec. 2018 (N = 56)	Jan. to Dec. 2019 (N = 253)	Jan. to Dec. 2020 (N = 87)	Jan. to Dec. 2021 (N = 132)
On admission	3.6%	15.4%	13.8%	22.0%
During hospitalization	29.6%	25.5%	16.7%	24.5%

## Data Availability

Not applicable.

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
