# Peer review of "Active Surveillance for Carbapenem-Resistant Enterobacterales (CRE) Colonization and Clinical Course of CRE Colonization among Hospitalized Patients at a University Hospital in Thailand"

_antibiotics, 2022, doi:10.3390/antibiotics11101401_

Round 1
Reviewer 1 Report
Dear Authors,
I have read the paper titled "Active surveillance for CRE Colonization and clinical course of CRE Colonizitation among hospitalized patients at a University Hospital in Thailand " with interest.
The topic is of interest, material and methods are sound, results well explained.
I have some questions.
Patients and Methods
a) Line 74
"CRE were detected using chomogenic selective agar"
Do you have confirmed the isolation of these CRE by determination of Carbapenem Mic ?
How did you interpret the results? Please specify
b) Line 78
You have identified the CRE species, by " conventional biochemical testing"
Please specify which biochemical testing were performed.
c) line 133 Karnofsky score
Please add the reference
Author Response
Reviewer 1
I have some questions.
Patients and Methods
- a) Line 74
"CRE were detected using chromogenic selective agar"
Do you have confirmed the isolation of these CRE by determination of Carbapenem Mic?
How did you interpret the results? Please specify\
Response
The method section of the manuscript said that “CRE was detected in the collected stool or rectal swab samples using chromogenic selective agar (CHROMagar mSuperCARBA, CHROMagar Company, Paris, France)”. Therefore, we did not perform carbapenem MIC to determine CRE. We used chromogenic selective agar (CHROMagar mSuperCARBA, CHROMagar Company, Paris, France) which was reported to be accurate for detecting CRE from stool samples said in these 2 references 1) García-Fernández S, Hernández-García M, Valverde A, Ruiz-Garbajosa P, Morosini MI, Cantón R. CHROMagar mSuperCARBA performance in carbapenem-resistant Enterobacteriaceae isolates characterized at molecular level and routine surveillance rectal swab specimens. Diagn Microbiol Infect Dis. 2017;87(3):207-9., and 2) Soria Segarra C, Larrea Vera G, Berrezueta Jara M, Arévalo Mendez M, Cujilema P, Serrano Lino M, et al. Utility of CHROMagar mSuperCARBA for surveillance cultures of carbapenemase-producing Enterobacteriaceae. New Microbes New Infect. 2018;26:42-8.
However, our laboratory has performed a validation of agreement between conventional culture and antibiotic susceptibility testing for detection of CRE and this chromogenic selective agar in 120 stool or rectal swab samples and we found that the test was accurate before we used such selective agar to detect CRE in our laboratory. Therefore, chromogenic selective agar (CHROMagar mSuperCARBA, CHROMagar Company, Paris, France) has been used for detection of CRE in stool sample in our laboratory over the past 5 years.
Interpretation of the results of this chromogenic selective agar was performed according to the company’s manual. If there are pink colonies of bacteria grown on the chromogenic selective agar plate containing the stool sample after incubation of the inoculated agar plate at 35C overnight, they are usually carbapenem-resistant E. coli. If there are dark blue colonies of bacteria grown on the chromogenic selective agar plate containing the stool sample after incubation of the inoculated agar plate at 35C overnight, they are usually carbapenem-resistant non- E. coli.
The references said in the above paragraph were added in the revised manuscript.
- b) Line 78
You have identified the CRE species, by " conventional biochemical testing"
Please specify which biochemical testing were performed.
Response
Identification of CRE species was performed by manual biochemical tests, including the triple sugar iron test, lysine iron agar slants test, indole test, motility test, ornithine decarboxylase test, urease test, and malonate test, which were locally prepared in the laboratory using the purchased reagents/materials (Oxoid Ltd.; Hampshire, UK or BBL/Difco Diagnostic, Becton Dickinson; Sparks, MD, USA), and the oxidase test (BBL/Difco Diagnostic, Becton Dickinson; Sparks, MD, USA), to identify E. coli and Klebsiella species.
The aforementioned statement was added in the method section of the revised manuscript.
- c) line 133 Karnofsky score
Please add the reference
Response
The following reference was added in the revised manuscript.
Péus D, Newcomb N, Hofer S. Appraisal of the Karnofsky Performance Status and proposal of a simple algorithmic system for its evaluation. BMC Med Inform Decis Mak. 2013 Jul 19;13:72.
Reviewer 2 Report
Major comment.
The strength of this paper is that it points out the high prevalence of CRE on admission in 1 Thai hospital, and also the high risk of acquitting CRE during hospitalization. These findings are important evidence supporting the need for better infection control measures at this hospital.
The weakness is that I found many parts of this paper confusing, particularly the sections about duration of CRE carriage.
Unless significant changes are made, I would recommend only including data and conclusions about prevalence on admission and acquisitions during hospitalization.
For example, line 120 states, "The median duration of persistent CRE colonization was 10 days (range, 1–151 days)." This median seems very short, compared to other studies. For example, in a meta-analysis (Bar-Yoseph H, et al. J Antimicrob Chemother. 2016 ) 74.6% of CRE carriers were still colonized 3 months after CRE detection. The authors state that the median length of stay for CRE carriers (positive on admission) was 12.5 days. If the median duration of CRE carriage was 10 days, how could 84.5% of patients still be positive at the time of discharge (line 120)?
Line 154 states, "The median duration of CRE colonization from the first detection of CRE to hospital discharge was 8 days (range, 1–49 days)." I didn't understand what this is measuring. Are you saying there were a median of 8 days from CRE detection to hospital discharge? That is not the same as duration of CRE colonization (unless everyone was negative for CRE at the time of discharge.) My same question applies to line 187. Later on, in the paragraph on CRE colonization after hospital discharge, I found that only 41 patients continued follow-up after discharge. If there was so little follow-up, how can the authors say that median duration of carriage was 10 or 8 or 6 days?
Specific comments
Abstract
1) Incidence should be reported as X cases per 1000 patient-days. If you do not have these data, then instead of using the term "incidence," say "28.3% of patients who tested negative for CRE on admission acquired CRE during their hospitalization." (This comment also applies to the results section.)
2) "Colonization was associated with healthcare exposures." Do you mean being colonized at the time of admission was associated with healthcare exposures? (Everyone who acquired CRE in the hospital had a healthcare exposure.)
3) "the incidence of CRE infection was 4.7%" This is not a correct of the term "incidence." You can say, "among CRE-colonized patients, 4.7% developed a CRE clinical infection."
4) "CRE colonization among hospitalized patients at Siriraj Hospital is prevalent." I think you mean there is a high prevalence.
Introduction
1) "Community spreading of CRE from patients with CRE colonization and infection is also a serious concern"7 Reference 7 is about ESBL-producing Enterobacterales, not CRE. You should cite a reference about community transmission of CRE.
2) page 2, line 45 "should be very helpful." Helpful for what? Reducing morbidity and mortality?
Methods
1) page 2, line 72. Please indicate who participated in the follow-up after hospital discharge. All patients enrolled in the study? Only those who were positive on admission or acquired CRE during hospitalization?
2) page 2 – "Multivariate analysis was performed using a backward stepwise binary logistic regression method." Please state what the outcome and main predictor variable was for this regression.
Results
1) All tables – I recommend that you remove the bullet points and align the variables column to the right side, instead of in the center. For example:
Comorbid conditions
Hypertension
Diabetes
2) Line 106 – "the overall prevalence of CRE colonization in their stool or rectal swab samples was 15.5%." Does this refer to prevalence on admission, or the percentage of patients who were positive on admission plus patients who acquired CRE during hospitalization?
3) Table 2 includes all the data from Table 1. There is no need for both. Delete Table 1.
4) I recommend changing the order of the results. First, present the data about CRE prevalence on admission, including Tables 2 and 3. Then present the data about CRE acquisition during hospitalization. Then present the data about follow-up after discharge.
5) line 124 – " The median duration of CRE colonization in this group of patients was 8 days (range, 4–51 days)." It's not clear who this group is.
6) Line 157 "The median CRE colonization from the first detection of CRE to the absence of CRE in the patients with CRE colonization upon admission was 6 days (range, 4–26 days)." I am confused about the different durations of colonization: 10 days (line 120), 8 days (line 124), 6 days (line 157). Please make it clear which patients each median refers to.
7) Table 4. It would be clearer to call this table "Comparison of 233 patients with and without CRE acquisition during hospitalization."
8) When referring to patients who were CRE negative on admission and acquired CRE during hospitalization, it would be clearer to say "patients who acquired CRE during hospitalization" rather than "patients with CRE colonization during hospitalization" (e.g. line 179).
Author Response
Reviewer 2
Major comment.
The strength of this paper is that it points out the high prevalence of CRE on admission in 1 Thai hospital, and also the high risk of acquitting CRE during hospitalization. These findings are important evidence supporting the need for better infection control measures at this hospital.
The weakness is that I found many parts of this paper confusing, particularly the sections about duration of CRE carriage.
Unless significant changes are made, I would recommend only including data and conclusions about prevalence on admission and acquisitions during hospitalization.
Responses
- We will try to correct the confusing part of the manuscript about duration of CRE carriage by reordering and removal of some parts of the duration of CRE carriage in the revised manuscript.
- The data in the abstract of the manuscript already said “The prevalence of CRE colonization upon admission was 15.5%, while the incidence of acquisition of CRE colonization during hospitalization was 28.3%.” Therefore, such data may not need to be mentioned again at conclusion of the abstract.
For example, line 120 states, "The median duration of persistent CRE colonization was 10 days (range, 1–151 days)." This median seems very short, compared to other studies. For example, in a meta-analysis (Bar-Yoseph H, et al. J Antimicrob Chemother. 2016) 74.6% of CRE carriers were still colonized 3 months after CRE detection. The authors state that the median length of stay for CRE carriers (positive on admission) was 12.5 days. If the median duration of CRE carriage was 10 days, how could 84.5% of patients still be positive at the time of discharge (line 120)?
Line 154 states, "The median duration of CRE colonization from the first detection of CRE to hospital discharge was 8 days (range, 1–49 days)." I didn't understand what this is measuring. Are you saying there were a median of 8 days from CRE detection to hospital discharge? That is not the same as duration of CRE colonization (unless everyone was negative for CRE at the time of discharge.) My same question applies to line 187. Later on, in the paragraph on CRE colonization after hospital discharge, I found that only 41 patients continued follow-up after discharge. If there was so little follow-up, how can the authors say that median duration of carriage was 10 or 8 or 6 days?
Response
This study was observational study and the decision to discharge the patients was made by their responsible physicians. Therefore, the investigators could not involve in such decision. Moreover, variation of the duration of hospitalization of the enrolled patients was very wide and the investigators could detect CRE in stool samples of only the patients who were in hospital. So, and median duration of CRE carriage in hospital seemed to be short because it was limited by the duration of hospitalization and detection of CRE in stool samples was performed every week while the patients were hospitalized.
We presented the data on median durations for both groups of the enrolled patients with persistent CRE colonization and with an absence of CRE colonization upon admission. For the patients with CRE colonization upon admission, the median duration of persistent CRE colonization from the first detection of CRE to hospital discharge was 8 days (range, 1–49 days) whereas the median duration of CRE colonization from the first detection of CRE to an absence of CRE in the patients with CRE colonization upon admission was 6 days (range, 4–26 days).
For the patients with CRE acquisition during hospitalization, the median duration of CRE colonization from the first detection of CRE to hospital discharge was 12 days (range, 5–52 days) whereas the median duration of CRE colonization from the first detection of CRE to an absence of CRE in the patients with CRE acquisition during hospitalization was 8 days (range, 5–51 days).
Specific comments
Abstract
1) Incidence should be reported as X cases per 1000 patient-days. If you do not have these data, then instead of using the term "incidence," say "28.3% of patients who tested negative for CRE on admission acquired CRE during their hospitalization." (This comment also applies to the results section.)
Response
We have revised the manuscript according to your suggestion.
2) "Colonization was associated with healthcare exposures." Do you mean being colonized at the time of admission was associated with healthcare exposures? (Everyone who acquired CRE in the hospital had a healthcare exposure.)
Response
We have changed the phase you mentioned to “CRE colonization upon admission was usually associated with prior healthcare exposure.”
3) "the incidence of CRE infection was 4.7%" This is not a correct of the term "incidence." You can say, "among CRE-colonized patients, 4.7% developed a CRE clinical infection."
Response
We have revised the manuscript according to your suggestion.
4) "CRE colonization among hospitalized patients at Siriraj Hospital is prevalent." I think you mean there is a high prevalence.
Response
We have changed the statement you said to“The prevalence of CRE colonization among hospitalized patients at Siriraj Hospital is high.”
Introduction
1) "Community spreading of CRE from patients with CRE colonization and infection is also a serious concern"7 Reference 7 is about ESBL-producing Enterobacterales, not CRE. You should cite a reference about community transmission of CRE.
Response
The reference for CRE in community was changed to “Kelly AM, Mathema B, Larson EL. Carbapenem-resistant Enterobacteriaceae in the community: a scoping review. Int J Antimicrob Agents. 2017 Aug;50(2):127-134.”
2) page 2, line 45 "should be very helpful." Helpful for what? Reducing morbidity and mortality?
Response
We have changed the phrase you said to “Moreover, selecting appropriate antibiotics to treat patients with CRE colonization who develop subsequent CRE infection should be very helpful for reducing morbidity and mortality.”
Methods
1) page 2, line 72. Please indicate who participated in the follow-up after hospital discharge. All patients enrolled in the study? Only those who were positive on admission or acquired CRE during hospitalization?
Response
We have revised the relevant statement to “All survived participating patients and/or their relatives at hospital discharge were asked and instructed to collect stool samples at home every 1 to 3 months and sent the samples to the investigator for detection of CRE in stool.”
2) page 2 – "Multivariate analysis was performed using a backward stepwise binary logistic regression method." Please state what the outcome and main predictor variable was for this regression.
Response
We have changed the statement you mentioned to“Multivariate analysis was performed using a backward stepwise binary logistic regression method with CRE colonization as the dependent variable and the potential risk factors, such as previous use of antibiotics, previous CRE colonization or infection, duration of hospitalization for the development of CRE colonization as the independent variables.”
Results
1) All tables – I recommend that you remove the bullet points and align the variables column to the right side, instead of in the center. For example:
Comorbid conditions
Hypertension
Diabetes
Response
The bullet points were removed and the alignments of the texts were adjusted.
2) Line 106 – "the overall prevalence of CRE colonization in their stool or rectal swab samples was 15.5%." Does this refer to prevalence on admission, or the percentage of patients who were positive on admission plus patients who acquired CRE during hospitalization?
Response
We have changed the statement you mentioned to “The prevalence of CRE colonization upon admission to hospital in their stool or rectal swab samples was 15.5%."
3) Table 2 includes all the data from Table 1. There is no need for both. Delete Table 1.
Response
Table 1 was deleted according to your suggestion.
4) I recommend changing the order of the results. First, present the data about CRE prevalence on admission, including Tables 2 and 3. Then present the data about CRE acquisition during hospitalization. Then present the data about follow-up after discharge.
Response
The order of the results was changed according to your suggestions.
5) line 124 – " The median duration of CRE colonization in this group of patients was 8 days (range, 4–51 days)." It's not clear who this group is.
Response
The above sentence was removed to avoid unclear description.
6) Line 157 "The median CRE colonization from the first detection of CRE to the absence of CRE in the patients with CRE colonization upon admission was 6 days (range, 4–26 days)." I am confused about the different durations of colonization: 10 days (line 120), 8 days (line 124), 6 days (line 157). Please make it clear which patients each median refers to.
Response
This study was observational study and the decision to discharge the patients was made by their responsible physicians. Therefore, the investigators could not involve in such decision. Moreover, variation of the duration of hospitalization of the enrolled patients was very wide and the investigators could detect CRE in stool samples of only the patients who were in hospital. So, and median duration of CRE carriage in hospital seemed to be short because it was limited by the duration of hospitalizationand detection of CRE in stool samples was performed every week while the patients were hospitalized.
We presented the data on median durations for both groups of the enrolled patients with persistent CRE colonization and with an absence of CRE colonization upon admission. For the patients with CRE colonization upon admission, the median duration of persistent CRE colonization from the first detection of CRE to hospital discharge was 8 days (range, 1–49 days) whereas the median duration of CRE colonization from the first detection of CRE to an absence of CRE in the patients with CRE colonization upon admission was 6 days (range, 4–26 days).
For the patients with CRE acquisition during hospitalization, the median duration of CRE colonization from the first detection of CRE to hospital discharge was 12 days (range, 5–52 days) whereas the median duration of CRE colonization from the first detection of CRE to an absence of CRE in the patients with CRE acquisition during hospitalization was 8 days (range, 5–51 days).
7) Table 4. It would be clearer to call this table "Comparison of 233 patients with and without CRE acquisition during hospitalization."
Response
The title of this table was changed according to your suggestion.
8) When referring to patients who were CRE negative on admission and acquired CRE during hospitalization, it would be clearer to say "patients who acquired CRE during hospitalization" rather than "patients with CRE colonization during hospitalization" (e.g. line 179).
Response
These wordings in the manuscript were changed according to your suggestion.
Reviewer 3 Report
This paper is the result of an active surveillance of CRE carriers at a university hospital in Thailand.
The fact that approximately 30% of patients contracted CRE in the hospital gives the strong impression that CRE is not endemic in this healthcare facility, but that this is simply an extremely dangerous healthcare facility for patients with uncontrolled nosocomial infections.
On the other hand, as an educational role model, this paper may be a particularly good report. But there are several other issues that should be discussed.
1. Authors need to explain the background of CRE infected patients in this study and the relationship between CRE carriers and CRE infected patients.
2. The 60% mortality rate of CRE infected patients is very high. The authors need to include a description of the background factors of CRE-infected patients who died and an explanation of the causes of the high mortality rate.
3. The authors need to further describe the analysis of how CRE carriers became CRE-infected patients and died.
4. What is the season and timing of CRE isolation? 3 years is a long period of time, so the authors need to explain the distribution over time in figures, etc.
5. What is the distribution of drug susceptibility among CRE isolates? This result is also very important to understand the content of this paper and needs to be explained in figures and tables.
6. The authors state that the 528 new patients were randomly selected, but there is a possibility that the selection method is biased based on these results. The method and rationale should be described.
If the hospital does not have nosocomial infection control measures in place, the actual CRE retention rate may be much higher.
7. If the authors claim that the frequency of CRE carriage in general internal medicine is higher than in other departments, they should provide specific data rather than speculative statements in the discussion section. If the hospital does not have nosocomial infection control measures in place, the reviewer thinks that differences among departments do not exist.
Finally, the authors' revisions must be reflected in the body of the paper.
Author Response
Reviewer 3
- Authors need to explain the background of CRE infected patients in this study and the relationship between CRE carriers and CRE infected patients.
Response
The explanation for the issue that you mentioned was added to the result section of the revised manuscript.
“All patients with CRE infection had prolonged course of current or previous hospitalization. The median duration from the first detection of CRE colonization in stool or rectal swab to CRE infection of these patients was 15 days.”
- The 60% mortality rate of CRE infected patients is very high. The authors need to include a description of the background factors of CRE-infected patients who died and an explanation of the causes of the high mortality rate.
Response
All patients with CRE-infection had multiple comorbidities. Two of them died from the complication of their comorbidities which was not directly related to CRE infection. We have added the aforementioned statements in the revised manuscript. However, our previous study about the natural course of CRE infection found that the mortality rate of patients with CRE infection in our center was 47.7%. We have added this information in the revised manuscript that this high mortality rate was due to limited treatment options in our center.
- The authors need to further describe the analysis of how CRE carriers became CRE-infected patients and died.
Response
The further descriptionof the analysis of how CRE carriers became CRE-infected patients and died was added as follow:
“Subsequent CRE infections following CRE colonization by the same CRE phenotypic bacteria have been observed in previous studies, especially among neutropenic patients, other immunocompromised patients or critically-ill patients. Translocation of bacteria from intestine to blood is considered a potential mechanism.”
- What is the season and timing of CRE isolation? 3 years is a long period of time, so the authors need to explain the distribution over time in figures, etc.
Response
We would like to add one more table instead of figure as shown below:
Table 5. Prevalence of CRE fecal carriage of the enrolled patients upon admission and during hospitalization
|
|
Nov. to Dec. 2018 (N=83) |
Jan. to Dec. 2019 (N=363) |
Jan. to Dec. 2020 (N=126) |
Jan. to Dec. 2021 (N=189) |
|
On admission |
3.6% |
15.4% |
13.8% |
22.0% |
|
During hospitalization |
29.6% |
25.5% |
16.7% |
24.5% |
The prevalence of CRE colonization in stool samples of the enrolled patients in 2018, 2019, 2020, and 2021 are shown in Table 5. There was a trend of increasing the prevalence of CRE colonization over 4 years of the study except that in 2020 when pandemic COVID-19 occurred and the number of the patients was small because it was quite difficult to enroll the patients during COVID-19 pandemic. Although COVID-19 was still epidemic in our hospital in 2021, the severity of the disease was much lower than that in 2021 and most of the patients and health personnel had received COVID-19 vaccination. Therefore, the number of the enrolled patients in 2021 was more than that in 2020.
- What is the distribution of drug susceptibility among CRE isolates? This result is also very important to understand the content of this paper and needs to be explained in figures and tables.
Response
We performed active surveillance for CRE detection by using chromogenic selective agar. This technique was chosen because it is simple, accurate, and fast and we were able to identify patient who had CRE colonization within 24 hours. Unfortunately, antibiotic susceptibility for all CRE strains isolated from stool or rectal swab samples was not performed in this study. However, for the patients who developed subsequent CRE infection after CRE colonization, 60% of these CRE isolates were susceptible to colistin which was used as the main therapy of CRE infection in combination with other agents because almost all CRE isolates in our hospital produced Metallo-beta-lactamases and other antibiotics that were active against such CRE with Metallo-beta-lactamases, such as cefiderocol is not available in Thailand.
- The authors state that the 528 new patients were randomly selected, but there is a possibility that the selection method is biased based on these results. The method and rationale should be described.
If the hospital does not have nosocomial infection control measures in place, the actual CRE retention rate may be much higher.
Response
Our hospital has nosocomial infection control measures, but it is not implemented in the patient without documented CRE. We did active surveillanceof CREin order to detect the patients with CRE fecal carriage so that isolation precaution will be implemented to these patients. Otherwise, these patients will not have isolation precaution. As mentioned in the manuscript that there were many hospitalized patients on each day and we did not have sufficient resources, especially the personnel to include all of them into the study. Therefore, we had to collect the patients on only some random days which we believed that such measure should not have bias. We have added more detail on the selection method as follow:
“Adult patients admitted to general medicine wards on randomly selected surveillance days were enrolled. The once-weekly random day was scheduled because the study team had limited time and resources to enroll all patients.”
- If the authors claim that the frequency of CRE carriage in general internal medicine is higher than in other departments, they should provide specific data rather than speculative statements in the discussion section. If the hospital does not have nosocomial infection control measures in place, the reviewer thinks that differences among departments do not exist.
Response
We revised the manuscript as follow:
“It should be emphasized that the high prevalence of CRE colonization of the patients upon admission was observed in individuals admitted to general medicine wards. The data on the prevalence of CRE colonization in patients admitted to other types of wards (e.g., surgery, obstetrics and gynecology, and ophthalmology) are not available. However, it is speculated that the prevalence of CRE fecal carriage in the patients hospitalized to other departments should be less than that of Department of Medicine because those patients have less healthcare-associated conditions, such as having chronic underlying diseases, prior use of antibiotics, receipt of antibiotics while hospitalization. Therefore, the study on active surveillance of CRE in the patients hospitalized to other departments in addition to Medicine Department should be done to consider if active surveillance of CRE should be performed in all hospitalized patients or only some patients in some departments.”
Finally, the authors' revisions must be reflected in the body of the paper.
Response
Thank you, we have revised the manuscript according to most of the reviewers’ suggestions.
Round 2
Reviewer 3 Report
No further comments.